# A Narrative Review of Alternative Symptomatic Treatments for Herpes Simplex Virus

**DOI:** 10.3390/v15061314

**Published:** 2023-06-02

**Authors:** Jane Y. Chang, Curt Balch, Joseph Puccio, Hyung S. Oh

**Affiliations:** 1Ascendant Biotech Inc., Foster City, CA 94404, USA; 2Bioscience Advising, Ann Arbor, MI 48103, USA; 3Department of Pediatrics, University of South Florida, Tampa, FL 33602, USA; 4Department of Microbiology, Blavatnik Institute, Harvard Medical School, Boston, MA 02115, USA; hyungsuk_oh@hms.harvard.edu

**Keywords:** herpes simplex virus, lysine, *Melissa officinalis*, nucleoside analogs, propolis

## Abstract

Herpes simplex virus-1 (HSV-1) and -2 (HSV-2) are large, spherically shaped, double-stranded DNA viruses that coevolved with *Homo sapiens* for over 300,000 years, having developed numerous immunoevasive mechanisms to survive the lifetime of their human host. Although in the continued absence of an acceptable prophylactic and therapeutic vaccine, approved pharmacologics (e.g., nucleoside analogs) hold benefit against viral outbreaks, while resistance and toxicity limit their universal application. Against these shortcomings, there is a long history of proven and unproven home remedies. With the breadth of purported alternative therapies, patients are exposed to risk of harm without proper information. Here, we examined the shortcomings of the current gold standard HSV therapy, acyclovir, and described several natural products that demonstrated promise in controlling HSV infection, including lemon balm, lysine, propolis, vitamin E, and zinc, while arginine, cannabis, and many other recreational drugs are detrimental. Based on this literature, we offered recommendations regarding the use of such natural products and their further investigation.

## 1. Introduction

The World Health Organization (WHO) estimates that over 4 billion people are infected with herpes simplex virus (HSV) types-1 (HSV-1, 3.7 billion cases) and -2 (HSV-2, 400 million cases) [1]. Infection by both HSV-1 and -2 causes lifelong clinical manifestations, including cold sores, genital ulceration, and encephalitis, and is the leading cause of corneal blindness in the U.S. [2]. Although the characteristic common cold sores are not life-threatening, infection of neonates or immunocompromised individuals can severely disseminate. Despite longstanding attempts at therapy and prevention, HSV remains among the most prevalent human infectious viral pathogens. Despite nearly 100 years of attempts to develop a vaccine, none were successful [3].

Consequently, due to the lack of effective prophylaxis and therapy, HSV-infected patients sought a variety of treatment options for their diseases, including “alternative” antiviral medications. However, patients seeking alternative therapies are exposed to the risks of harm from misinformation. Therefore, the demand for proper public information cannot be ignored. In this review, we summarize the breadth of alternatives to established and experimental HSV pharmaceuticals.

## 2. Methods

Two authors (JYC and CB) conducted a literature search of EMBASE, Medline, and Google Scholar for the terms “natural products and HSV” or “alternative treatments to HSV” and independently screened abstracts for clinically relevant articles. Disagreements on article inclusion of specific agents were discussed before incorporation into the review, in line with historical support and pertinence.

## 3. HSV Pathogenesis

HSV-1 and HSV-2 infect orofacial and genital mucosal surfaces, respectively, while sharing a common structure, consisting of a large (>84 genes), double-stranded, linear DNA genome, encased within a capsid (i.e., “nucleocapsid”), contained within a lipid bilayer envelope. The nucleocapsid is tethered to the envelope by “teguments,” structural, virally encoded proteins unique to HSV, with the complete particle known as a virion (Figure 1) [4].

As shown in Figure 1, HSV infection occurs in several discrete steps. Virion attachment occurs via HSV glycoproteins [4], and the nucleocapsid is then transported along microtubules to the nucleus, wherein viral DNA is released for replication. Transcription of HSV genes is catalyzed the host’s RNA polymerase II [4,5], with early genes encoding enzymes involved in DNA replication and biosynthesis of envelope glycoproteins, and late genes predominantly encoding proteins that form the virion particle, ultimately resulting in host cell egress of mature virions (Figure 1) [5].

Productive infection forms vesicular lesions in the mucosal epithelia, followed by spread of the virus to sensory neurons and establishment of a latent infection that typically remains for the life of the host [5]. Reactivation of dormant virus results in recurrent disease at or adjacent to the site of primary infection.

### 3.1. Established HSV Treatments: Antivirals

Acyclovir is an established standard treatment for HSV infection, with therapeutic benefits for both oral and genital herpes. Acyclovir is a 9-(2-hydroxyethoxymethyl) guanine, an acyclic nucleoside analog highly selective against HSV-1, HSV-2, and varicella-zoster virus (VZV) [6]. While the human host thymidine kinase phosphorylates the nucleoside thymidine, herpes thymidine kinase phosphorylates human host guanine and its analog, acyclovir, to create acyclovir monophosphate, which is subsequently phosphorylated to acyclovir diphosphate and triphosphate. Because acyclovir lacks ribose, a 5-ring sugar with a hydroxyl group required for DNA polymerase elongation, the nascent viral DNA chain is terminated. Furthermore, acyclovir competitively inhibits and inactivates the HSV DNA polymerase [6].

Acyclovir can be administered intravenously, orally, or topically. Although it is most effective when administered early in primary infections, it can also serve as a long-term prophylactic in suppressing recurrence and reducing symptoms of latent viral reactivation [7]. Intravenous acyclovir is a viable option for herpes simplex encephalitis in adults, and it was successful in treating disseminated HSV in pregnant women and neonates [7]. However, acyclovir is not known to protect against post-herpetic neuralgia.

Valacyclovir and valganciclovir are the prodrugs—inactive drugs biologically activated in vivo—of acyclovir and ganciclovir, respectively, and are also used against HSV infection [8]. These, however, also can produce adverse events such as neutropenia, anemia, and thrombocytopenia [9].

### 3.2. Resistance

Acyclovir resistance is generally caused by HSV mutations in either its thymidine kinase (95% of acyclovir-resistant isolates) or DNA polymerase genes, resulting in decreased or absent HSV thymidine kinase production and loss of DNA polymerase affinity for acyclovir-triphosphate [10]. Resistance is also described primarily in immunocompromised patients with a clinical presentation of chronic ulcerative mucocutaneous disease, with prolonged shedding of the virus, but also rarely in immunocompetent patients [11].

Early studies in 1988 determined that based on dosage, long-term suppressive acyclovir therapy (1 year) was safe and efficacious for recurrent genital herpes, with no critical side effects observed [12], while another study showed no viral resistance even after 6 years of acyclovir use [13].

However, more recently, evidence that contradicts prior resistance studies emerged. With an increasing number of immunocompromised patients at a much greater risk of resistance, acyclovir-resistant HSV strains rose as well [14]. Immunocompromised patients exhibit a stable prevalence of 4–7% [15], compared to 0.3% in immunocompetent individuals [16], while others observed a surprisingly high incidence (6.4%) of acyclovir-resistant strains in immunocompetent herpetic keratitis patients [17]. Further studies suggest that acyclovir-resistant viruses are much more likely to cause recurrent corneal disease than other diseases such as herpes labialis or herpes genitalis [16]. Although foscarnet and cidofovir showed benefit against acyclovir-resistant HSV isolates, these are primarily used against cytomegalovirus, generally require intravenous administration, and can elicit significant cytotoxicity [18].

Evidence suggests that long-term acyclovir prophylaxis is an important protective factor against latency re-emergence of acyclovir-resistant HSV active isolates [19,20]. However, resistance can lead to treatment-refractory disease and, subsequently, worse clinical outcomes, with immunocompromised patients at an even higher risk of complications. Hence, new safe and effective antiherpetic compounds, with novel mechanisms of action, are urgently required.

## 4. Supplements and Natural Products

Since HSV is an incurable, lifelong disease with discomfiting symptoms, in addition to limited approved pharmaceuticals, it has a long history of proven and unproven (or even harmful) home remedies, many of which are now disseminated on the Internet. Unproven and likely unbeneficial remedies include hot/cold compresses, cornstarch paste, numerous essential oils, baking soda, and sitz baths. Likewise, Luminance RED lacks peer-review support of efficacy (merely testimonials), although the benefits of myriad Chinese and other herbs (used for thousands of years) now formally showed efficacy in vitro [21,22]. Likewise, extracellular polymeric substances (EPSs), including various polyanions and sulfated polysaccharides (secreted by various microorganisms), likewise demonstrated in vitro antiviral efficacy, including blockage of virion ingress and immune activation [23]. Here, we examined some other well-known natural products with lower cytotoxicity profiles than synthetic drugs [24], which have some evidence of benefit or detriment to HSV infection (Table 1). While various mechanistic studies were performed in cell culture, we endeavored, whenever possible, to correlate these to human or in vivo animal studies.

### 4.1. Lysine/Arginine

Lysine is an essential amino acid that humans cannot synthesize but must acquire through food intake, particularly dairy and meat, and for vegetarians and vegans, via supplements. Lysine is well known for relieving the symptoms of HSV, as decades of research support that high intracellular levels of lysine have inhibitory effects on HSV multiplication in cell cultures [25].

Clinical research also supported the success of lysine in reducing the recurrence of HSV infection. A 6-month-long double-blind, multicenter experiment using oral L-lysine monohydrochloride showed that the lysine treatment group averaged 2.4 times fewer HSV outbreaks, with significantly diminished symptoms and shorter healing times (during the research period) than the placebo group [26]. A similar study examining lysine prophylactic usage for HSV suppression concluded that a significantly higher number of patients were recurrence-free, on lysine vs. placebo [26].

Lysine appears to be a successful daily prophylactic agent for reducing HSV recurrence, and is an effective therapeutic agent for decreasing the severity and healing time for HSV reactivation. However, a review of the literature reported that <1 g/day lysine supplementation was ineffective for HSV prophylaxis or lesion treatment, while doses >3 g/day improved patients’ subjective disease experience [26]. Even so, when compared to the established antiviral treatment, acyclovir, lysine is a safe natural compound, without reported adverse side effects, and is a promising alternative treatment option for patients with HSV.

While the exact mechanism of action of lysine against HSV is not fully known, it is well-established that lysine acts as an antimetabolite that suppresses the HSV viral growth-promoting actions of its analog, arginine [25]. Arginine is found in high-protein white meats (such as chicken, pork, and turkey) and legumes. Unlike lysine, arginine exhibits the opposite desired effects on HSV, i.e., promotion of HSV viral growth, acting as a medium for growth of HSV that is essential for viral replication [27]. Specifically, it was found that in arginine-deficient cell medium, HSV was rendered incapable of replicating; replication immediately resumed, however, when arginine-deficient culture was exchanged for arginine-rich medium [28]. A threshold concentration of arginine was found for HSV growth, and virus yield increased with increased arginine concentration [24].

**Table 1 viruses-15-01314-t001:** Possible benefits and detriments of various natural products to the pathology of herpes simplex virus.

Natural Substance	Hypothesized Mechanism of Action	Clinical Studies?	Results	Adverse Events	References
Arginine	Amino acid, enhances viral protein synthesis	Multiple studies	Exacerbated disease	Increased viral replication	[25,28,29]
Cannabis	Possible membrane perturbation, may enhance viral DNA synthesis, immunosuppressive	Multiple observational studies	Likely exacerbated disease, enhanced HSV replication, increased viral shedding	Anxiety, potential psychoses, long-term cognitive decline	[30,31,32]
Lemon balm	Prevents virus ingress	Phase II and preclinical	Reduced symptom scores in patients; in vitro, shows highly reduced cell death and infection	No known adverse effects	[33,34]
Lysine	Amino acid, attenuates viral protein synthesis	Multiple	Decreased outbreak severity and duration; prophylactic	Safe up to 6 g per day	[35]
Refined carbohydrates	Multiple	Phase I	Exacerbated disease	Decreased innate immunity	[36,37]
Vitamin C	Antioxidant, boosts immune response	Limited	57% reduced time to remission; reduced HSV keratitis recurrence by 53.2%	Minor effects at very high doses	[38,39]
Vitamin D	Largely unknown, but may inhibit Toll-like receptor detrimental inflammation	Observational	Reduced viral titers in a HeLa cells HSV- infection model, reduced Behçet’s disease in an HSV-1 mouse model	Hypercalcemia or GI effects with high doses	[40,41]
Vitamin E	Immunomodulation	Preclinical	Alleviated mouse herpes simplex encephalitis	Bleeding at excessively high doses	[42]
Zinc (oral)	Antioxidant, direct inhibition of HSV DNA polymerase	Multiple	Reduced number and duration of lesions	Excessive doses (100–300 mg) may elicit copper deficiency	[43]
Zinc (topical)	Antioxidant, direct inhibition of HSV DNA polymerase	Multiple	Shortened lesion outbreaks, decreased complications	No observed adverse events	[44,45]

Therefore, in vitro data may be the basis for the conclusion that patients prone to HSV recurrence should abstain from excess arginine, particularly during periods of stress, and should acquire supplemental lysine in their diet. Educating patients with the correct dosage, especially finding the correct composition of arginine, is also very important for patients who are currently suffering or examining the use of lysine as a treatment option. Therefore, patients infected with HSV and other viruses should refrain from consuming high amounts of arginine, particularly during reactivation. However, the health benefits of numerous high arginine foods, including legumes, nuts, turkey/chicken breast, and unrefined grains (e.g., oats, whole wheat, brown rice), should also be weighed when considering arginine abstinence.

### 4.2. Propolis

Propolis, a resin-like material synthesized by bees to coat small openings in their hives, is a natural product widely used in home remedies, with many proposed pharmacological properties, including antiviral, anti-inflammatory, antibacterial, antioxidant, anticancer, antiglycemic, and antifungal effects [46]. Propolis is produced from the combination of bee discharge, beeswax, and tree sap [47]. While propolis composition varies depending on the type of trees and flowers approached by bees, geography, and season of collection, raw propolis comprises approximately 50 percent resins, 30 percent beeswax and fatty acids, 10 percent essential oils, 5 percent pollen, and 5 percent of various organic compounds such as vitamins, minerals, and sugars [48].

However, the most important complex chemical compound that renders propolis an antiviral agent is phenolic acid. It was also proposed that benzoic acid, galangin, caffeic acid, pinocembrin, and chrysin may be effective active ingredients against HSV in cell culture [49].

While successful mechanisms of action studies remain mostly lacking [48], it was hypothesized that propolis may negatively affect HSV-1 and HSV-2 viral replication in host cells [47]. For example, Yildrim et al. (2016) effectively showed antiherpetic viral replication by Hatay propolis [47]. It was observed that viral replication and cytopathological changes, such as virus aggregation, rounding of cells, and nuclear enlargement, were inhibited in both HSV-1 and HSV-2 cell cultures treated with propolis [47]. Combined with acyclovir, propolis inhibited HSV-1 replication after 24 h of incubation, and HSV-2 replication after 48 h of incubation, with both viruses showing suppressed viral DNA and mRNA levels. A study examining the in vitro anti-HSV-1 activity of 3-methyl-but-2-enyl caffeate, a constituent of propolis, supported its suppression of HSV DNA/RNA levels. This constituent was also shown to reduce HSV-1 viral titers by 3log_10_, and viral DNA synthesis, by 32-fold [50]. A similar study examining the effects of propolis extract ACF^®^ (PPE) suggested that PPE had noticeable adsorption, and virucidal effects, against both HSV-1 and HSV-2 [51].

The antiviral properties of propolis are readily apparent in clinical settings. For example, a randomized, double-blind, placebo-controlled clinical trial was conducted to assess the efficacy of 3% propolis ointment ACF (Herstat) in patients with a history of HSV-1 infection. Patients treated with propolis ointment had a mean healing time of 6.24 days, compared to 9.77 days in the placebo group, in addition to earlier pain relief. Overall, 81.8% of the surveyed propolis ointment group judged their treatment to be ‘very effective’, and 18.2% ‘somewhat effective’, compared to 60% of the placebo group finding the ointment ‘hardly effective’ and 22.9% ‘ineffective’ [52].

Currently, while propolis remains underutilized, it has vast potential for the symptomatic treatment of herpes. With increasing support that propolis’s viral replication inhibition is comparable to that of acyclovir, it is safe to conclude that propolis can suppress HSV-1 and HSV-2 viral recrudescence. Furthermore, it is noteworthy to consider utilizing propolis in conjunction with acyclovir, to synergistically suppress HSV-1/-2 recurrence over acyclovir alone [48].

### 4.3. Lemon Balm

Another widely regarded beneficial anti-HSV agent is lemon balm (*Melissa officinalis* L.), a traditional herbal medicine accepted as a mild sedative, spasmolytic, and antibacterial agent [53]. In one study, a hydroalcoholic lemon balm leaf extract reduced HSV-2 cytotoxicity, by 60%, at a nontoxic concentration of 0.5 mg/mL, via inhibiting virion entry into Vero kidney cells [33]. Similarly, a *Melissa* extract was found to be 80% and 96% inhibitory, respectively, of cellular attachment by acyclovir-sensitive and -resistant HSV-1 strains [54].

In humans, a double-blind study of 66 patients with recurrent herpes labialis demonstrated that a lemon balm cream (leaf extract) significantly reduced symptom scores (in blistered areas), after two days of treatment, compared to a placebo cream [55]. Likewise, it was found in another randomized, double-blind study of 116 patients, that lemon balm application, initiated within 72 h of symptom onset, two to four times daily for ten days, significantly increased self-reported healing in 41% of treated respondents, vs. 19% of those receiving a placebo cream [56].

### 4.4. Vitamin E

Vitamin E (α-tocopherol) is a well-known fat-soluble antioxidant, believed by many to enhance immune function [57,58]. With regard to HSV, weanling male BALB/cByJ mice, fed a four-week vitamin E-deficient diet prior to intranasal challenge with HSV-1, exhibited severe symptoms of encephalitis, keratitis, hunched posture, and morbidity, compared to those pre-fed a diet with adequate vitamin E [42]. A follow-up to this study demonstrated that vitamin E-deficient mice had fewer CD8+IFN-γ+ T cells, and activated dendritic cells, trafficking into the brain from the periphery, in addition to systemically increased T regulatory cells (Tregs) [59].

In a study of a topical formulation, dried and cotton saturated with vitamin E oil (25,000 units) was placed over ulcerative herpetic gingivostomatitic lesions for 15 min, resulting in lesion regression and pain relief, within eight hours [60]. In another study of 50 patients with herpetic cold sores, vitamin E application to lesions, every four hours, resulted in prompt and sustained pain relief, and the lesions healed more rapidly than expected [61].

### 4.5. Zinc

Zinc, an essential trace element for humans, is a cofactor for over 300 enzymes, including the antioxidant superoxide dismutase [62]. Over two billion people, worldwide, are affected by zinc deficiency, which associates with numerous diseases, including childhood developmental disorders, and was well-linked to dysfunction of both humoral and cell-mediated immunity, increasing susceptibility to infection [63].

In an in vitro study of ten randomly selected HSV-1 and HSV-2 clinical isolates, nine were > 97% inactivated by pretreatment with zinc lactate, in plaque assays of CV-1 monkey kidney cells [64]. Aside from free virus inactivation, zinc may inhibit various other steps in the HSV life cycle, including protein synthesis [65] and DNA replication (polymerase inhibition) [66], although the latter effect was disputed [67].

In a study of 20 recurrent herpes labialis patients systemically treated twice daily (oral) with 22.5 mg zinc sulphate, for four months over a one-year period, a significant reduction in herpetic lesion episodes, from six to three, was observed [43]. Likewise, ten recurrent genital herpes patients receiving daily 50 mg zinc sulfate showed a 57% reduction in the total number of days with lesions present, with greater reductions in lesion frequency with each successive month of treatment (thus, unlikely related to a placebo effect) [68]. Interestingly, deficient human serum zinc levels significantly positively correlated with duration of recurrent herpes labialis lesions (i.e., delayed recovery) [69].

Zinc also demonstrated anti-HSV efficacy when applied topically. In 46 participants with facial or circumoral herpes, 21-day application of a zinc oxide/glycine cream significantly reduced duration of cold sores and severity of symptoms (blistering, soreness, itching, tingling), compared to a placebo cream [44]. In a separate study of herpes genitalis, only 1% of patients (receiving 4% ZnSO_4_, in distilled water) and 6% of patients (receiving 2% ZnSO_4_) recurred, after a six-month treatment, compared to 80% of those given a distilled water control; moreover, no side effects were observed [45]. More recently, a formulation of polyethylene glycol-coated zinc oxide nanoparticles, at 200 µg/mL, 2.5 log_10_ reduced an HSV-1 50% tissue culture infectious dose (TCID_50_), while also inhibiting HSV-1 genomic DNA copy number by 92% [70].

### 4.6. Vitamin D

Vitamin D is a family of lipid-soluble steroid hormones synthesized in the lower epidermis through a photochemical reaction with ultraviolet (UVB) light, while dietary sources include milk and various types of fish [71]. In particular, the vitamin D3 prohormone is hydroxylated in the kidneys, to form the bioactive agents 25-hydroxyvitamin D3 (25(OH)D3) and 1,25 dihydroxyvitamin D3 (1,25(OH)_2_D3) [71]. Both of these activated hormones significantly downregulated HSV-1 titers in infected HeLa cells, in addition to repressing mRNA expression of Toll-like receptors [40], inflammatory mediators previously shown as host-detrimental in HSV infection [72,73].

In another study, in a mouse model of HSV-induced Behçet’s disease (BD), oral administration of 1,25(OH)_2_D3 ameliorated BD symptoms in six of eleven mice, while also downregulating the Toll-like receptors TL2 and TL4 [41]. In an analysis of human data from 14,174 participants in the National Health and Nutrition Survey, from 2007 to 2016, it was found that vitamin D deficiency was a risk factor for both HSV-1 and HSV-2 (odds ratios of 2.205 and 2.704, respectively) [74].

### 4.7. Refined Carbohydrates

While little-studied in-depth, it is widely recognized that chocolate and other sweets can exacerbate HSV outbreaks [36]. In one rat study, the addition of sucrose to the diet (10–20% of energy) caused a dose-dependent reduction in the capacity to produce antibodies [37], while an in vitro model of intestinal lumen and blood demonstrated that fructose downregulated phagocytosis mediated by mannose-binding lectin, an innate immune response pattern recognition molecule, against influenza A-infected Madin-Darby (MDCK) cells [75]. Likewise, a ketogenic (high fat/low carbohydrate) diet was found to protect against weight loss and neuroinflammation of HSV-1-induced herpes simplex encephalitis, in a mouse model of that condition, via alterations in the gut microbiome [76].

In human studies, it was shown that acute hyperglycemia impaired IL-6 and IL-17 expression, particularly in CD14+CD16+ intermediate monocytes [77]. More recently, it is widely believed that consumption of a “Western diet” (i.e., high in refined carbohydrates and saturated fats) contributed to both incidence and mortality during the COVID-19 pandemic [78], while recent support for a link between HSV and type II diabetes is also noteworthy [79].

### 4.8. Cannabis

*Cannabis* is a genus of flowering plants widely used for medicinal and recreational purposes, containing over 80 chemical constituents known as cannabinoids, and over 200 terpenes. The two most researched cannabinoids are Δ9-tetrahydrocannabinol (Δ9-THC), cannabis’s primary psychoactive component, and cannabidiol (CBD), a nonpsychoactive component with many hypothesized medical benefits, such as reduced inflammation, antibacterial activity, and analgesia [30]. Numerous studies concluded that THC enhances HSV viral replication [31]. For example, pretreatment of HSV-2-infected cells with micromolar concentrations of Δ9-THC increased the number of infectious virions by 100-fold [80], while others showed that Δ9-THC enhances HSV-2 virion egress, by perturbing the host cell’s plasma membrane [31]. Cabral et al. also conducted a study in which guinea pigs were administered Δ9-THC, prior to intravaginal introduction of HSV, demonstrating significantly more severe genital disease, higher mean vagina-shed virus titers, and higher mortality, during a 30-day study period, compared to controls, implying that Δ9-THC decreases guinea pig resistance to HSV-2 vaginal infection [32]. Δ9-THC and cannabis extracts were also shown to suppress the killing capacity of cytotoxic T and natural killer (NK) cells, against HSV- and other virally infected cells, from THC-treated mouse spleens [81,82]. Likewise, more recent reviews found that cannabis is immunosuppressive in users, and it likely impairs immune responses against human immunodeficiency virus (HIV) and hepatitis C virus (HCV) [83,84].

Contrary to these studies, other studies found conflicting evidence of THC’s inhibitory effects of THC on HSV-infected human cells. For example, it was found that neither HSV-1 or HSV-2 were replicative or extensively cytopathic to human cell monolayer cultures, where different concentrations of Δ9-THC were introduced before, during, and after HSV infection [31]. THC-mediated inhibition of HSV replication was both time- and dose-dependent, and not influenced by the pH of the culture medium, thus suggesting that THC preferentially reduces the infectivity of enveloped HSV [85].

Besides THC, other cannabis constituents may provide beneficial antiviral effects. The other primary cannabinoid, CBD, has only weak affinity for the endocannabinoid receptors CB1 and CB2, and exhibits distinct bioactivity. One study found that CBD reduced proliferation of, and induced apoptosis in, Kaposi sarcoma-associated HSV-infected endothelial cells [86], while CBD also inhibited hepatitis C replication in an in vitro model [87]. Various other studies suggested that terpenes (numbering over 200 in cannabis) possess specific antiviral properties, including the virion ingress that initiates host cell infection [88,89].

One clear application of medical marijuana in HSV treatment is that it can help patients cope with HSV. The cyclical incurable prognosis of HSV leads to psychosocial issues such as stress, anxiety, and depression [3], leading to a positive cycle of heightened pain sensation with HSV recurrence, and cannabinoid treatment can ease HSV-induced neuropathic pain [30]. However, considering its possible immunosuppressive activity, one should weigh the risks and benefits of cannabis for its analgesic and anxiolytic properties. Some studies claimed no effect on HSV replication, and it is clear that medical marijuana lacks unifying evidence of contradictions against HSV. Given its conflicting evidence, the use of whole cannabis products (or their full extracts), by HSV-infected individuals, should be given discretion.

### 4.9. Other Addictive/Recreational Substances

Similar to cannabis, most other addictive or recreational drugs, including opiates, cocaine, nicotine, and alcohol, are likely detrimental to HSV etiology and exacerbation. For example, acute morphine administration to BALB/c mice significantly reduced cytotoxic T lymphocyte (CTL) responses, lymphocyte proliferation, and IFN-γ production, contributing to reactivation of latent HSV-1 [90]. Similarly, adoptive transfer of spleen cells, from morphine (vs. saline)-treated BALB/c mice, to HSV-1-infected, cyclophosphamide (an immunomodulator)-treated mice, significantly increased mortality, via suppression of IFN-γ production and natural killer (NK) cell activity [91].

Similar to mouse studies, in one cross-sectional retrospective survey of 70 sexually active young adults (non-sex workers), the concurrent use of four to eleven possible illicit drugs was found to be a significant risk factor for genital herpes [92]. Another retrospective study of obstetric patients found that 9.7% of those receiving epidural opiates (morphine and/or fentanyl) developed recurrent oral herpes lesions, compared to only 0.6% of those not provided opioids (*p* < 0.001) [93]. With regard to alcohol overuse, a western Pennsylvanian study found HSV-2 seroprevalence, in sexually active female adolescents with alcohol use disorder (AUD), was twice the rate of those without AUD (males with AUD were not significantly affected) [94].

Immunomodulatory effects of psychedelics, another class of recreational, hallucinogenic drugs currently under study for numerous psychiatric disorders, remain unestablished. One early in vitro study showed that 100 µM lysergic acid diethylamide (LSD) suppressed proliferation of B lymphocytes, NK response, and production of anti-inflammatory cytokines; however, such effects were not seen in humans [95]. In one study, several classic psychedelics (LSD, psilocybin, N,N-dimethyltryptamine (DMT), and mescaline) were found nonaffective toward CTL activity and cytokine release [96]. Since many of these agents act on various 5-HT receptor subtypes [95], which are also expressed on leukocytes, their immunomodulatory effects should continue to be examined, in conjunction with these substances’ potential benefits for schizophrenia, post-traumatic stress disorder, and other serious mental disorders.

## 5. Conclusions

For 100 years, a permanent solution to HSV was the goal of scientific and medical innovation. There yet remains no effective treatment that can bring the human race a functional cure and a perpetual stop to this viral infection. In addition, the current gold standard treatment for herpes simplex virus, acyclovir, raises an important but overlooked issue of resistance and ineffectiveness with increased usage, leading to a need for research into new antiviral drugs and vaccines [97]. However, the path to the development of a new drug can be considerably costly, lengthy, and daunting. Based on these prior studies, we submit that propolis and 3–5 g lysine daily would be beneficial against recurrent HSV outbreaks, with additional potential prophylactic efficacy. Despite the efficacy of numerous natural products (including herbs), Web-based claims of HSV “cures”, based on cocktails of these substances, remain unsubstantiated, anecdotal, and peer-unreviewed. We would, likewise, submit prudence in refraining from cannabis and most illicit substances, high arginine-to-lysine foods, and refined carbohydrates. Taken together, the identification and research of more efficacious compounds will combat the direct and indirect effects of this pervasive scourge to the human race.

## Figures and Tables

**Figure 1 viruses-15-01314-f001:**
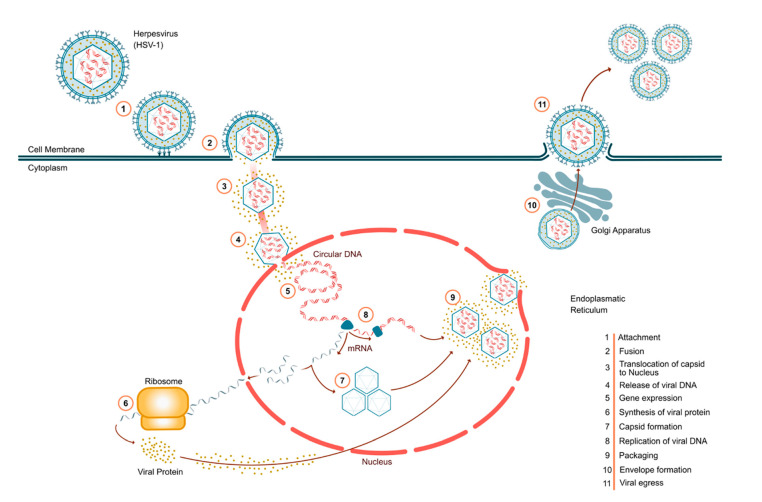
The 11 steps in the life cycle of herpes simplex viruses. Virion attachment (ingress) occurs via specific glycoproteins (1), which also elicit membrane fusion (2). The capsid is then transported to the nucleus (3) and viral DNA released (4) and expressed (5). Viral transcripts are then translated to protein at ribosomes (6), allowing reformation of capsids (7). Meanwhile, viral DNA replication (8) occurs, which is then packaged into capsids (9). Upon budding from the Golgi, the envelope is reformed to encase capsids (10), followed by egress of progeny virions (11). While not firmly established, the beneficial natural products discussed herein likely interfere with protein synthesis (lysine, arginine, steps 6,7) DNA replication (propolis, step 8), and egress (terpenes, step 11).

## Data Availability

Not applicable.

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
