# Peer review of "A Narrative Review of Alternative Symptomatic Treatments for Herpes Simplex Virus"

_viruses, 2023, doi:10.3390/v15061314_

Round 1
Reviewer 1 Report
Please, attached you will find my coments.

Reviewer 2 Report
In the manuscript entitled “A Narrative Review of Alternative Symptomatic Treatments for Herpes Simplex Virus”, the authors compared the treatment effects of a few natural products to acyclovir and suggested potential long-term prophylactic HSV treatments in place of acyclovir. The review will be of moderate interests to herpes virologists. Several issues need to be addressed before a potential publication.
Major points:
1. Reference #10 is a review, which has been repeatedly referred to in Section 3. Try not to rephrase statements/conclusions from published reviews as the main body of a new review. It is better to use primary literature to elaborate the authors’ opinion and direct the readers to see other review(s) if those contents have already been reviewed.
2. Page 4, section 3: “Based on this evidence, we conclude that long-term acyclovir prophylaxis is an important protective factor against the emergence of acyclovir-resistant HSV.” Not sure what this means. Doesn’t long term usage induce drug resistance?
3. Table 1: beneficial or detrimental to host or virus? In the text, it is said “…have some evidence of benefit or detriment to HSV infection (Table 1).” However in table 1, the results that are termed “detrimental” seemingly refer to harmful effects on the host, not the virus.
4. Table 1: add a column of references to include a complete list of literature relevant to the table.
5. Many inhibitory effects of the natural products introduced in this review rely on cell culture. The authors should explain the limitations of cell culture effects and speculate the effects of these natural products on host cell growth when discussing their potential applications as the prophylactic treatments.
Minor points:
1. On page 2, section 3: Genome is encased in the capsid. The encased genome along with the capsid is called nucleocapsid.
2. Page 4: “high intracellular levels of lysine’s inhibitory effects on HSV multiplication in cell cultures” should be changed to “that high intracellular levels of lysine have inhibitory effects on HSV multiplication in cell cultures”
3. Line 26: “may affect HSV-1 and HSV-2 viral cell division”. Did you mean division of infected cells or viral replication in infected cells?
Extensive English editing is needed. I only marked a few places.
Round 2
Reviewer 1 Report
I consider that the manuscript can be accepted in this revised version.